# The community acceptance of COVID-19 vaccines in Rakhine State: A cross-sectional study in Myanmar

Saw Simon[1]*, Kaung Myat Min[2], Tun Zaw Latt[3], Pa Pa Moe[4], Kyaw Myo Tun[5]

**1** Faculty of Health and Social Sciences, University of Bedfordshire, Luton, United Kingdom, **2** University Research Co., LLC (URC), Eliminate Malaria Burma Project, Yangon, Myanmar, **3** Myanmar Health Assistant Association, Yangon, Myanmar, **4** Myanmar Nurse and Midwife Association, Yangon, Myanmar, **5** Department of Health and Social Sciences, STI Myanmar University, Yangon, Myanmar

* dr.simon.grace@gmail.com

**Data Availability Statement:** The datasets mentioned in this research are available in online repositories, and the repository names as well as the corresponding accession numbers are

## Abstract

The global pandemic situation of SARS-CoV-2 (COVID-19) has been ongoing for more than 2 years with the emergence of different variants. With the rapid development of vaccines, countries including Myanmar rolled out vaccination programs to reduce the morbidity and mortality due to COVID-19 with the ultimate goal to end the pandemic. This study seeks to explore the acceptance of the general adult population towards the COVID-19 vaccines administered by the Ministry of Health, and barriers to vaccine acceptance. A quantitative cross-sectional study was conducted by adopting valid and reliable questionnaires from similar studies around the world. Simple random sampling was used to select 288 participants from 12 townships of Rakhine State, Myanmar. The interview was performed using standardized paper-based documents. While the data entry and manipulation were performed using Microsoft Excel, the data analysis process was performed using the Statistical Package for Social Science (SPSS) software. As descriptive statistics, the level of vaccine acceptance, and barriers to vaccine acceptance were calculated. Chi-square analysis and bivariate logistics regression was performed to explore the associated socio-demographic characteristics, COVID-19 and vaccine-related experience, and perceptions of participants on the health belief model (HBM) domains related to vaccine acceptance. A total of 276 participants entered the study and revealed an overall vaccine acceptance level of 91.3%. Higher level of education, working in skilled manual and sales services, monthly income of more than 200,000 MMK (111 USD), history of previous vaccination, not experiencing side effects of vaccine after previous immunization, and elements of the health belief model (HBM) were associated with higher vaccine acceptance. The barriers to vaccine acceptance were mistrust of the efficacy of vaccines and potential major adverse events of COVID-19 vaccines. The high level of vaccine acceptance among the general population in Rakhine state provides an opportunity for health authorities to achieve high vaccination coverage within the community. Nevertheless, the vaccine-related education campaigns should be targeted and conveyed frequently to the sub-groups of the population with vaccine hesitancy to obtain the highest achievable level of vaccine coverage within the community for the ultimate goal to end the pandemic.

provided below: https://figshare.com/s/
61577bcefb92002f87cc.

**Funding:** The authors received no specific funding for this work.

**Competing interests:** The authors have declared that no competing interests exist.

## Introduction

Coronavirus disease 2019 (COVID-19) is a newly discovered infectious disease which was first identified during a pneumonia outbreak in Wuhan city of China [1, 2]. When the curative treatment for the infection was not available, the World Health Organization (WHO) encouraged the nations to initiate, adapt or raise public health and social measures (PHSM) depending on the intensity of the transmission and capacity of the national health system [3].

With the advance in research and technology, vaccine development for COVID-19 has been faster than usual, with multiple vaccine candidates undergoing clinical trials and 14 vaccines approved by WHO for emergency use after examining the quality, safety, and effectiveness of each vaccine [4, 5]. These vaccines use different technologies, including whole virus, protein subunit, nucleic acid, and viral vector [6]. The acceptance of COVID-19 vaccines is crucial to controlling the pandemic, however, concerns about safety, adverse effects, and efficacy are affecting the vaccine acceptance and hesitancy of the population [7, 8]

Since the community acceptance of COVID-19 vaccines is paramount for success of vaccination program with the goal to control the pandemic, several studies have been conducted to determine the acceptance of COVID-19 vaccines among the population in different countries. A literature review on vaccine acceptance rates of 114 global countries identified vaccine hesitancy is more prominent in the Middle East and North America, Europe and Central Asia, and Western/ Central Africa while the highest vaccine acceptance rates in Asia and the Pacific [9]. The vaccine acceptance rates in countries from Asia and the Pacific region was described in Table 1.

While Myanmar faced four outbreak episodes, with the third wave starting in July 2021 (Fig 1). This wave was characterized by a high prevalence of the Delta variant and low vaccination coverage, resulting in a significant increase in daily new cases and deaths. [10, 11]. As of 30[th] April 2022, the cumulative caseloads of 612,883 and 19,434 deaths among lab-confirmed cases were reported with a case fatality ratio of 3.17% [12].

The Ministry of Health (MOH) Myanmar rolled out the vaccination program in January 2021 with Covidshield/ Astrazeneca vaccines arriving from the COVAX facility [13]. While the extra workforce was required for COVID-19 care and vaccination services, the health system including vaccination service was weakened by the Civil Disobedience Movement (CDM)

**Table 1. COVID-19 vaccine acceptance rates in countries from Asia and the Pacific, adapted from the review study of Sallam, Al-Sanafi and Sallam (20).**

| Sr | Country | Assessment Date | Acceptance Rate |
|---|---|---|---|
| 1 | Afghanistan | December 2020 –January 2021 | 63% |
| 2 | Australia | August 2020 | 59% |
| 3 | Bangladesh | January-February 2021 | 61% |
| 4 | China | January-February 2021 | 82% |
| 5 | Hong Kong | December 2020 –January 2021 | 42% |
| 6 | India | January 2021 | 79% |
| 7 | Indonesia | September 2020 | 65% |
| 8 | Japan | February 2021 | 56% |
| 9 | Malaysia | December 2020 | 83% |
| 10 | Nepal | December 2020 | 97% |
| 11 | New Zealand | March 2021 | 70% |
| 12 | Pakistan | January-February 2021 | 72% |
| 13 | Philippines | January 2021 | 63% |
| 14 | South Korea | May-June 2021 | 77% |
| 15 | Taiwan | October 2020 | 53% |
| 16 | Vietnam | October-December 2020 | 97% |

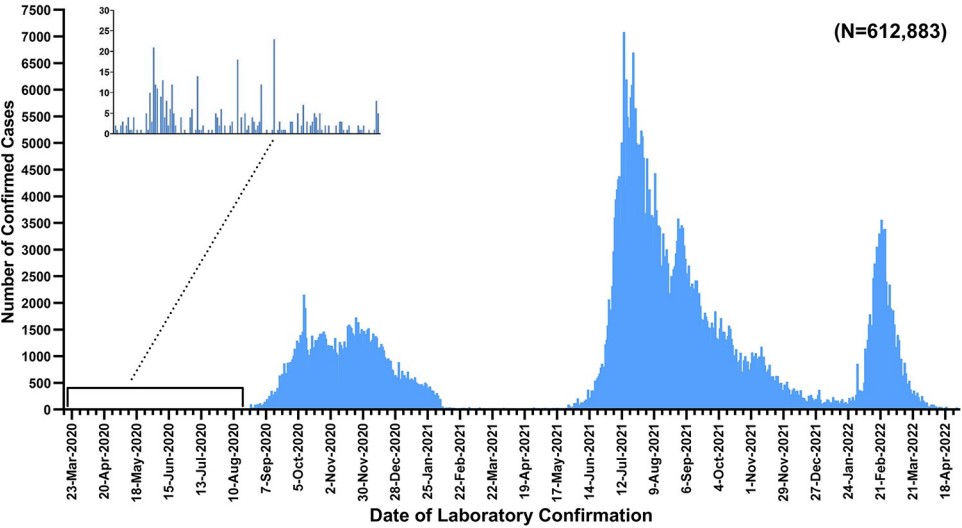

**Fig 1. Epidemic curve with different waves of COVID-19 epidemic in Myanmar (Data source: Ministry of Health, Myanmar).**

of healthcare workers and the vaccination plan was interrupted due to the military coup of 1st February 2021 [14]. Vaccination progress was accelerated again in October 2021 with vaccines from People's Republic of China, Russia, India and production of Myancopharm vaccines by Myanmar [15–19]. The summary of COVID-19 vaccines available in Myanmar was summarized in Table 2. As of April 30th 2022, 43.2% of the total population were fully vaccinated against COVID-19 in Myanmar where the vaccine coverage is the lowest among Southeast Asia countries [10, 20]. Although different vaccines were introduced to the local population, no prior studies are available to measure the vaccine acceptance of citizens which is vital for the vaccination program to obtain the highest attainable level of immunization coverage. Hence, the current study aimed to investigate the acceptance of the population towards the COVID-19 vaccines administered by the MOH and barriers to vaccine acceptance through community survey in selected region of Myanmar.

## Materials and methods

### Research design

Since this was a primary research study which aims to identify the level of acceptance of the community over the COVID-19 vaccines and further explore the associated background

**Table 2. Currently available COVID-19 vaccines in Myanmar classified by manufactured country and provider.** Source: Ministry of Health, Myanmar [13, 15–18].

| Sr | Name of Vaccine | Country of Origin | Vaccination Provider |
|---|---|---|---|
| 1 | Covishield/ AstraZeneca | India | MOH |
| 2 | Sinopharm | China | MOH |
| 3 | Sinovac/ Coronavac | China | MOH |
| 4 | Sputnik Light | Russia | MOH |
| 5 | Covaxin | India | MOH |
| 6 | Johnson & Johnson/ Janssen | United States | Private Sector (UN) |
| 7 | Moderna | United States | Private Sector (UN) |
| 8 | Myancopharm | Myanmar/ China | MOH |

characteristics, COVID-19-related perceptions and experiences, and potential barriers to vaccine acceptance at one point in time, a cross-sectional quantitative design was appropriate. Therefore, the current study has adopted the designs of former vaccine acceptance studies from Indonesia and Iraq to develop a valid set of questionnaires including socio-demographic characteristics and health belief model elements [21, 22]. The processes of the study were conducted from 1st February 2022 to 15th May 2022.

### Target population and sampling

**Target population.** Myanmar has seven states, seven regions and one union territory with a total population of 51.4 million as per the 2014 Myanmar Population and Housing Census report [23, 24]. To maximize the generalizability of the study, people across all regions of the country should be selected. However, with the limited human resources and timeframe of the study, it was not feasible to access citizens from every region of the country. The target population of the study was people living in 12 townships in Rakhine State of Myanmar Fig 2. Rakhine State is located on the western coast of Myanmar and connected the international land border with Bangladesh whereas different ethnic groups of Rakhine, Bengali and Chin people were living in the state [25]. Among the 12 townships of the study area, Maungdaw and Rathedaung townships are connected to the land border with Bangladesh where the frequent mass movement of migrants and displaced people across the border is uncontrollable [26, 27]. In addition, higher COVID-19 positive cases were reported in these two townships compared to other nearby townships within the state [28]. Therefore, conducting the current study including these townships covered different ethnic groups.

**Sample size.** Since there was no published study in Myanmar, the proportion of the acceptance of COVID-19 vaccines was estimated from the COVID-19 vaccines acceptance rates (67%-93.3%) of a study from Indonesia, one of the South-East Asia (SEA) countries with a similar context as Myanmar [21] and 88.6% of China, a neighbouring country of Myanmar [29]. Accordingly, the average proportion of the population accepting the COVID-19 vaccines in the current study was assumed 80%.

The sample size for estimating an infinite population proportion was computed from the formula provided in the published textbook of foundation for analysis in the health sciences [30].

$$n = \frac{z^2_{1-\frac{\alpha}{2}}p(1-p)}{d^2}$$

As per calculation with the above variables in the N4Studies mobile application, the sample size of 271 participants (246 + 25, sample size with adding 10% drop-out rate) was sufficient enough (Table 3). In this study, 288 participants were intended to interview for taking equal participants from the urban and rural populations of each township.

**Sampling procedure.** The current study mainly used probability sampling with face-to-face interviews to acquire reliable data from the participants and to generalize the research findings to the target population. A purposive sample was defined to achieve equal participants from the urban and rural areas of each township to reflect the vaccine acceptance in urban and rural populations. Then, simple random sampling was performed to randomly select 12 participants from households in the urban and rural areas with the sampling frame of COVID-19 response activity of University Research Co, LLC (URC). (i.e., Urban 12 household x rural 12 household x 12 townships = 288 participants).

**Eligibility criteria.** The eligible participants were household leaders, housewives or family members more than 18 years of age living in the targeted 12 townships of Rakhine State,

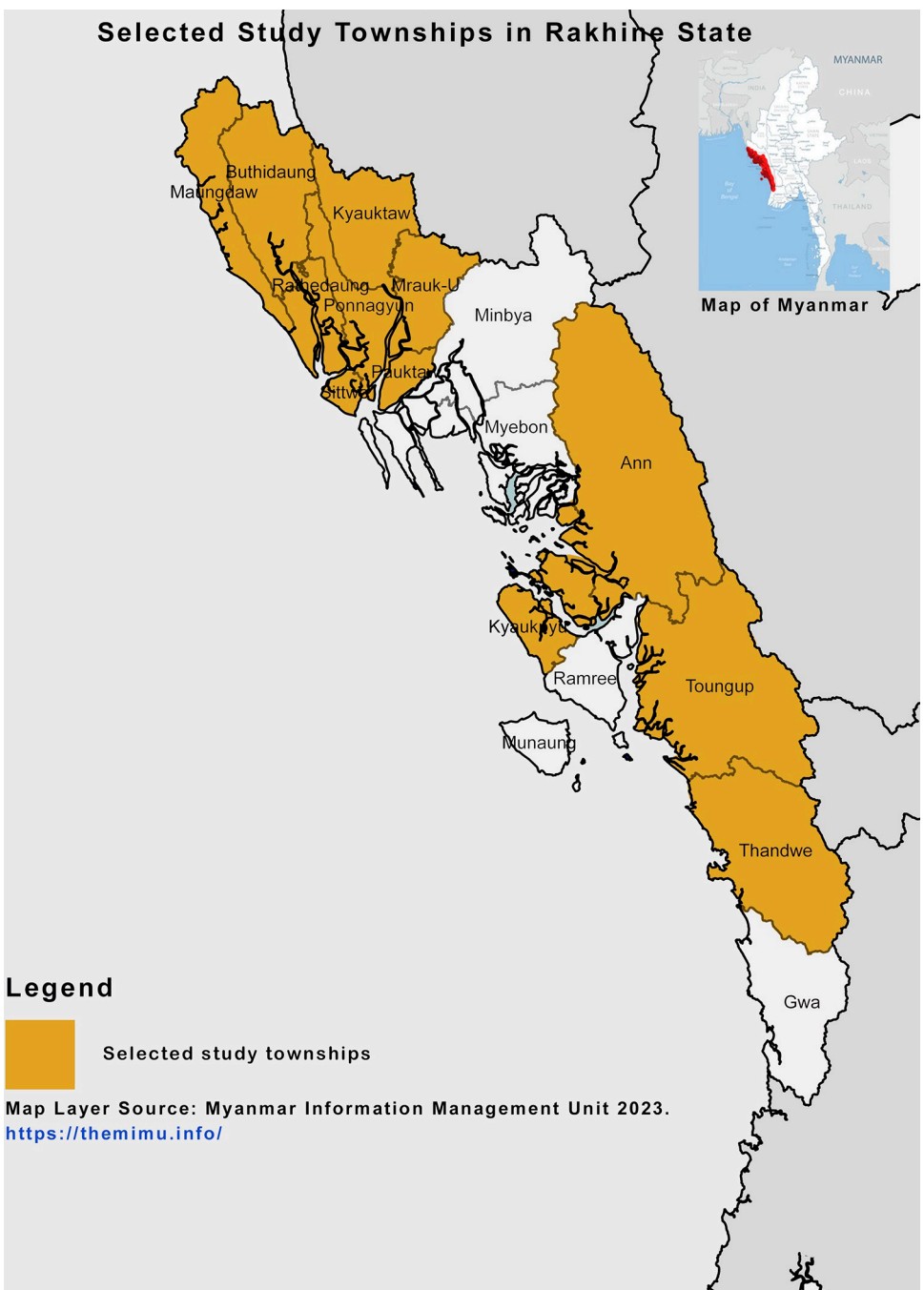

**Fig 2. Selected study townships within Rakhine State of Myanmar "Map Layer republished from the Myanmar Information Management Unit (MIMU) under a CC BY license, with permission from Mr. Naing Lin Kyaw, MIMU, 2023".**

Myanmar. From the randomly sampled household, the data collectors asked for the availability of a household leader with informed consent to participate in the study. The household leader is the individual who serves as the main breadwinner or primary source of income in a household. This person is typically identified as the male husband in Myanmar, although in cases where the husband has passed away, the role may be filled by a female housewife. In this study,

**Table 3. Sample size calculation for estimating infinite population proportion.**

| Outcome | Anticipated population proportion (P) | Absolute precision (d) | Level of Significance (α) | Sample Size (n) | Drop-Out Rate (10%) |
|---|---|---|---|---|---|
| Level of community acceptance of COVID-19 vaccines | 0.80 (80%) **Reference:** [7, 21] | 0.05 (5%) | 0.05 (95% CI) | 246 | 25 |

the household leaders were preferred to participate in the study as they could decide their household members to accept or refuse the vaccination. If the household leader was not accessible at the time of the interview, the housewife or household member more than 18 years of age were invited to participate in the study.

**Data collection tool.** As the data collection tool, the valid set of pre-existing questionnaires from similar studies around the world [7, 21, 22, 31–33] was reviewed to develop the set of questionnaires for this study including socio-demographic characteristics, experience with the COVID-19 infection and vaccination, HBM domains and reason for refusing the vaccines. The paper-based questionnaires were printed out and asked to the participants by the interviewer. For reliability (internal validity) analysis, the questionnaires were asked to 30 participants from different regions of the country and performed analysis by SPSS. As per the calculation provided by Field, the overall Cronbach's Alpha value was 0.91 [34].

**Ethical approval.** The study protocol was approved by Institutional Review Board of STI Myanmar University (STI IRB/03/22) and the Ethics Committee of the School Society Community and Health, University of Bedfordshire, UK (SSCHREC PUB010-6/STI011-6).

**Data entry and analysis.** The primary researcher used Microsoft Excel for data entry, verification, and manipulation. Cross-checking of electronic data with paper-based documents ensured data quality control. Statistical analyses were performed using SPSS Statistics software (v26, IBM Corporation). Descriptive statistics, chi-squared analysis, and bivariate logistic regression were conducted to assess community acceptance of vaccines and barriers to acceptance. Confidence intervals and p-values (p<0.05) were used to determine statistical significance.

## Results

### Socio-demographic characteristics of participants

Out of the targeted 288 participants, 276 were successfully interviewed from urban and rural areas of the 12 townships. Unfortunately, rural households from Toungup township could not be assessed. The participants' mean age (SD) in years was 41.56 (12.7) and they ranged in age from 18 to 80. The majority of the 276 study participants were men (51%), lived in cities (52%), had at least a high school education (40.6%), worked in sales and services (24.6%), made between 108,000 and 200,000 per month (43.1%), practiced Buddhism (89.9%), were married (85.9%), and were employed in other fields (93.8%). Table 4 provided a complete analysis of participants' sociodemographic traits.

### Community acceptance of COVID-19 vaccines

Regarding residence, 91.3% (n = 252) of participants accepted the COVID-19 vaccines provided by MOH to immunize themselves or their family members, with 8.7% (n = 24) expressing vaccine hesitation. The COVID-19 vaccination was accepted 100% in Toungup township, (95.8%) in Kyauktaw township, (91.7%) in Buthedaung, Maungdaw, Sittwe, Ponnagyun, Mrauk-U, Thandwe, Ann, and Kyaukpyu township, (87.5%) in Rathedaung township, and (83.3%) in Pauktaw township. The acceptance percentages for people between the ages of 18

**Table 4. Socio-demographic characteristics of participants (n = 276).**

| Socio-demographic characteristics | Number (n) | Percent % |
|---|---|---|
| **Age Group** | | |
| 18–30 | 56 | 20.3 |
| 31–40 | 86 | 31.2 |
| 41–50 | 68 | 24.6 |
| >50 | 66 | 23.9 |
| **Gender** | | |
| Male | 141 | 51.0 |
| Female | 135 | 49.0 |
| **Urbanicity** | | |
| Urban | 144 | 52.0 |
| Rural | 132 | 48.0 |
| **Education Level** | | |
| Illiterate/ Nonformal Education | 14 | 5.1 |
| Primary School | 23 | 8.3 |
| Middle School | 127 | 46.0 |
| High school and above | 112 | 40.6 |
| **Occupation** | | |
| Agriculture | 45 | 16.3 |
| Unskilled manual | 21 | 7.6 |
| Skilled manual | 66 | 23.9 |
| Sales and services | 68 | 24.6 |
| Clerical | 29 | 10.5 |
| Professional/ technical/ managerial | 28 | 10.1 |
| No occupation | 19 | 6.9 |
| **Income (in Myanmar Kyats)** | | |
| < 108,000 (60 USD) per month | 37 | 13.4 |
| 108,000–200,000 (60–111 USD) per month | 119 | 43.1 |
| > 200,000 (> 111 USD) per month | 101 | 36.6 |
| **Religion** | | |
| Buddhism | 248 | 89.9 |
| Islam | 24 | 8.7 |
| Christian | 4 | 1.4 |
| **Marital Status** | | |
| Single | 26 | 9.4 |
| Married | 237 | 85.9 |
| Divorced | 3 | 1.1 |
| Widow | 10 | 3.6 |
| **Health Related Sector** | | |
| No | 259 | 93.8 |

and 40 were almost comparable (87.2% vs. 87.5%), according to age group. The vaccine acceptability rate then increased linearly, starting at 94.1% in the 41–50 age range and reaching (97%) in those over 50, as shown in Fig 3.

## Factors associated with acceptance of COVID-19 vaccines

**Socio-demographic characteristics.** When identifying who was more likely to accept the COVID-19 vaccines according to participants' socio-demographic characters, the higher

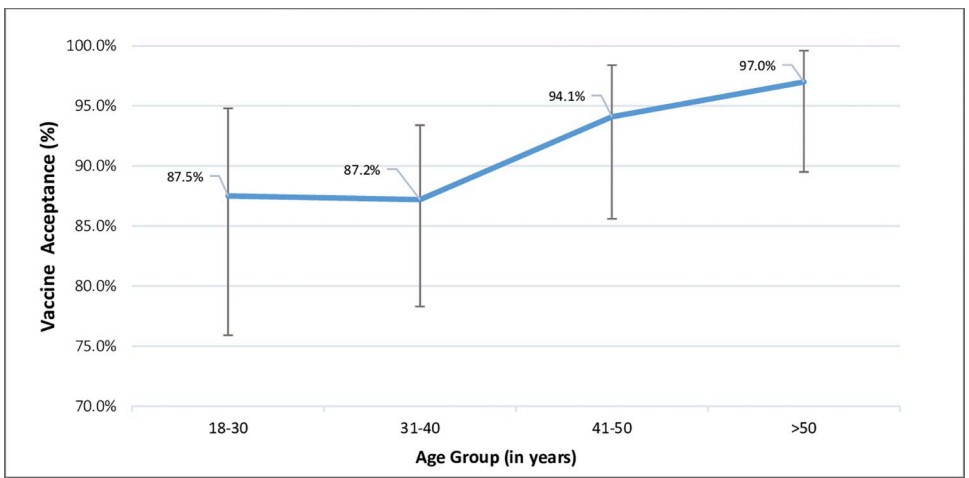

**Fig 3. Projected prevalence of COVID-19 vaccine acceptance (%) by age groups (years) in Myanmar adult population.**

vaccine acceptance proportions were found in over 50 years aged group (97.0%), males (92.9%), urban (94.4%), high-school and above education group (99.1%), clerical and professional/ technical/ managerial occupation group (100.0%), more than 200,000 MMK (111 USD) monthly income group (98.0%), singles (92.3%), and health related sector (91.5%) compared to their counterparts.

Although age, gender, urbanicity, religion, marital status, and employment status were not statistically linked to vaccine acceptability, significant relationships between vaccine acceptability and education ($P = 0.003$), occupation ($P = 0.001$), and monthly income ($P = 0.003$) were found respectively (Table 5).

**COVID-19 and vaccine-related experience.** Only 16 (5.8%) participants in this study had a prior history of COVID-19 infection in themselves or within their family members, of which, 8 (50%) of them experienced hospitalization and 3 (18.8%) experienced severe illness or death within their household members. In this study, 252 (91.3%) participants reported that they had heard about COVID-19 and its vaccines. A total of 276 participants, 267 (96.7%) were vaccinated alone or as a family. Of those, 47 (17.6%) reported vaccine-related side effects, whereas 220 (82.4%) did not. Eighty-five percent (n = 40) of the 47 participants who provided feedback on the appearance of vaccine side effects said they had experienced minor symptoms (such as fever, headache, muscle aches, and pain at the injection site), but no one reported major adverse events (MAE) related to the COVID-19 vaccine.

History of COVID-19 infection, hospitalization history, and experiences with severe sickness or death had no noticeable effects on vaccination uptake. However, the emergence of vaccine side effects following prior immunization was associated with consent to future immunization ($P = 0.034$), and prior immunization with COVID-19 was statistically associated with vaccine acceptance for subsequent immunization ($P = 0.001$) (Table 6).

**Elements of health belief model (HBM).** First, the percent likelihood that an infection will occur within the range of 0–20, 21–40, 41–60, and more than 60 were used to classify the perceived risk of infection. The majority of individuals (82.7%) thought their risk of infection was 40% or greater. And almost all of the participants (99.3%) obtained information about COVID-19 via the media (including newspapers, pamphlets, speakers in public, billboards, and social media), with the majority of those participants (85.4%) being more likely to accept the vaccination when the media promoted it. Surprisingly, no single participant disagreed with

**Table 5. The associations between socio-demographic characteristics and vaccine acceptance (n = 276).**

| Participants' factors | Willing to accept COVID-19 vaccines | | P-value |
| --- | --- | --- | --- |
| | Total n (%) | Acceptance n (%) | |
| *Socio-Demographic Characteristics* | | | |
| **Age Group** | | | 0.103 |
| 18–30 | 56 (20.3) | 49 (87.5) | |
| 31–40 | 86 (31.2) | 75 (87.2) | |
| 41–50 | 68 (24.6) | 64 (94.1) | |
| >50 | 66 (23.9) | 64 (97.0) | |
| **Gender** | | | 0.334 |
| Male | 141 (51.1) | 131 (92.9) | |
| Female | 135 (48.9) | 121 (89.6) | |
| **Urbanicity** | | | 0.053 |
| Urban | 144 (52.2) | 136 (94.4) | |
| Rural | 132 (47.8) | 116 (87.9) | |
| **Education Level** | | | < 0.001 |
| Primary School and below | 37 (13.4) | 16 (43.2) | |
| Middle School | 127 (46.0) | 125 (98.4) | |
| High school and above | 112 (40.6) | 111 (99.1) | |
| **Occupation** | | | < 0.00 |
| Agriculture & Unskilled manual | 66 (25.7) | 45 (68.2) | |
| Skilled manual & Sales services | 134 (52.1) | 132 (98.5) | |
| Clerical & Professional/ technical/ managerial | 57 (22.2) | 57 (100) | |
| **Income (in Myanmar Kyats)** | | | 0.003 |
| < 108,000 (60 USD) per month | 37 (13.4) | 30 (81.1) | |
| 108,000–200,000 (60–111 USD) per month | 119 (43.1) | 106 (88.2) | |
| > 200,000 (> 111 USD) per month | 101 (36.6) | 99 (98.0) | |
| **Religion** | | | 0.374 |
| Buddhism | 248 (89.9) | 226 (91.1) | |
| Islam | 24 (8.7) | 23 (75.0) | |
| Christian | 4 (1.4) | 3 (95.8) | |
| **Marital Status** | | | 0.675 |
| Single | 26 (9.4) | 24 (92.3) | |
| Married | 237 (85.9) | 217 (91.6) | |
| Divorced/ Widow | 13 (4.7) | 11 (91.3) | |
| **Health Related Sector** | | | 0.643 |
| No | 259 (93.8) | 237 (91.5) | |
| Yes | 17 (6.2) | 15 (88.2) | |

the assertion that "participants were more willing to take vaccines when health workers/MOH or WHO suggested immunization.

The Chi-square test found a highly significant association between the HBM components (perceived advantages, subjective norm, and cues to action) and the subject's acceptance of the vaccine (P<0.001). Table 7 displays all of the outcomes of the HBM domain analysis.

## Influencing factors for the acceptance of COVID-19 vaccines

Finding the influencing factors of participants' acceptance of the COVID-19 vaccines was done using logistic regression analysis (Table 7).

**Table 6. The association between COVID-19 and vaccine-related experience to the vaccine acceptance (n = 276).**

| Participants' factors | Willing to accept COVID-19 vaccines | | P-value |
|---|---|---|---|
| | Total n (%) | Acceptance n (%) | |
| *COVID-19 and vaccine-related experience* | | | |
| **History of COVID-19 infection in respondent or within the family** | | | 0.721 |
| No | 260 (94.2) | 237 (91.2) | |
| Yes | 16 (5.8) | 15 (93.8) | |
| **Hospitalization due to COVID-19 infection in respondent or within the family (n = 16)** | | | 0.302 |
| No | 8 (50.0) | 8 (100) | |
| Yes | 8 (50.0) | 7 (87.5) | |
| **Severe illness or death due to COVID-19 within the family (n = 16)** | | | 0.62 |
| No | 13 (81.3) | 12 (92.3) | |
| Yes | 3 (18.8) | 3 (100) | |
| **History of COVID-19 vaccination in respondent or within the family** | | | < 0.001 |
| No | 9 (3.3) | 5 (55.6) | |
| Yes | 267 (96.7) | 247 (92.5) | |
| **History of appearing side effects of COVID-19 vaccine in respondents or within the family (n = 267)** | | | 0.034 |
| No | 220 (82.4) | 207 (94.1) | |
| Yes | 47 (17.6) | 40 (85.1) | |
| **Type of side effects appeared (n = 47)** | | | N/A |
| Minor | 47 (100) | 40 (85.1) | |
| Major adverse events | 0 | 0 | |

**Socio-demographic characteristics.** Participants with a middle school education or higher were around 82.03 (95% CI = 17.57, 383.000) and 145.69 (95% CI = 18.32, 1158.51) times more likely to accept COVID-19 immunizations than participants with only primary education. In comparison to agriculture workers and unskilled manual workers, participants who worked as sales staff and skilled manual laborers were about 30.80 (95%CI = 6.95, 136.57) times more likely to acquire COVID-19 vaccinations.

**COVID-19 and vaccine-related experience.** Participants who had previously received the COVID-19 vaccination by themselves or by family members had a 9.88 (95% CI = 2.46, 39.73) fold higher chance of accepting the vaccine than those who had not. Participants who had previously received the COVID-19 vaccination and had no negative side effects had odds of vaccine acceptance that were 2.79 (95% CI = 1.05, 7.42) fold higher than those who had negative side effects (Table 8).

**Barriers to vaccine acceptance.** When 24 participants were asked an open-ended question about their reasons for refusing vaccination, 75% (n = 18) of the responses were due to mistrust of the effectiveness of the vaccinations, and 25% (n = 6) were related to possible serious side events from COVID-19 vaccines.

## Discussion

The study identified important new information relating to the area of COVID-19 vaccine acceptance in the Myanmar adult population. While the overall vaccine acceptance rate was 91.3% for the vaccines administered by MOH, the vaccine acceptance level upsurged with the increase in age reaching the highest acceptance rate of 97% in participants aged more than 50 years. The education level, type of occupation and income of the individual were associated with vaccine acceptance, but age, sex, urbanicity, religion, marital status, and working or studying in the healthcare-related sector were not associated. The previous history of COVID-

**Table 7. The association of health belief model elements and vaccine acceptance (n = 276).**

| Elements of Health Belief Model | Willing to accept COVID-19 vaccines | | (P-value) |
|---|---|---|---|
| | Total n (%) | Acceptance n (%) | |
| **Perceived risk of infection** | | | < 0.001 |
| 0–20% | 23 (8.3) | 7 (30.4) | |
| 21–40% | 20 (7.2) | 15 (75.0) | |
| 41–60% | 118 (42.8) | 115 (97.5) | |
| >60% | 110 (39.9) | 110 (100) | |
| Don't know | 5 (1.8) | 5 (100) | |
| **Perceived Benefits** | | | |
| **Feel safe and protected after the COVID-19 vaccination** | | | < 0.001 |
| Disagree | 7 (2.5) | 1 (14.3) | |
| Neutral | 36 (13.0) | 21 (58.3) | |
| Agree | 233 (84.4) | 230 (98.7) | |
| **Vaccines are effective in reducing COVID-19 infection** | | | < 0.001 |
| Disagree | 23 (8.3) | 1 (4.3) | |
| Neutral | 31 (11.2) | 8 (40.0) | |
| Agree | 222 (80.4) | 221 (95.7) | |
| **Vaccines are effective in reducing disease severity and mortality even infected with COVID-19** | | | < 0.001 |
| Disagree | 1 (0.4) | 0 (0) | |
| Neutral | 20 (7.2) | 8 (2.9) | |
| Agree | 255 (92.4) | 244 (88.4) | |
| **Subjective Norm** | | | |
| **More likely to accept vaccine after vaccination of friends** | | | < 0.001 |
| Disagree | 19 (6.9) | 2 (10.5) | |
| Neutral | 19 (6.9) | 15 (78.9) | |
| Agree | 238 (86.2) | 235 (98.7) | |
| **Cues to Action** | | | |
| **Ever heard about COVID-19 information from the media** | | | N/A |
| No | 2 (0.7) | | |
| Yes | 274 (99.3) | | |
| **More likely to accept vaccines when media recommend (n = 274)** | | | < 0.001 |
| Disagree | 19 (6.9) | 0 (0) | |
| Neutral | 21 (7.7) | 17 (81.0) | |
| Agree | 234 (85.4) | 233 (99.6) | |
| **More likely to accept vaccines when health workers/ MOH or WHO recommend** | | | < 0.001 |
| Neutral | 19 (6.9) | 5 (26.3) | |
| Agree | 257 (93.1) | 247 (96.1) | |

19 vaccination by the respondent or within the family and experience of side effects of COVID-19 vaccine in previous vaccination were influencing the willingness of respondents in accepting the vaccines. The HBM domains such as the perceived risk of infection, perceived benefits, subjective norms and cues to action were also associated with the vaccine acceptance of the population. When asking the participants who refused the vaccines, the doubt about the effectiveness of COVID-19 vaccines and the potential occurrence of major adverse events of vaccines were identified as the only two perceived barriers to vaccine acceptance in this study.

**Table 8. Logistic regression analysis to identify influencing factors of the acceptance of COVID-19 vaccines (n = 276).**

| Influencing factors | COR (95% CI) | p-value |
|---|---|---|
| **Socio-Demographic Factors** | | |
| **Age Group** | | |
| 18–30 | Reference | |
| 31–40 | 0.97 (0.35, 2.68) | 0.959 |
| 41–50 | 2.29 (0.63, 8.25) | 0.207 |
| >50 | 4.57 (0.91, 22.98) | 0.065 |
| **Gender** | | |
| Female | Reference | |
| Male | 1.52 (0.65, 3.54) | 1.516 |
| **Urbanicity** | | |
| Rural | Reference | |
| Urban | 2.35 (0.97, 5.68) | 0.059 |
| **Education Level** | | |
| Primary School and below | Reference | |
| Middle School | 82.03 (17.57, 383.00) | <0.001 |
| High school and above | 145.69 (18.32, 1158.51) | <0.001 |
| **Occupation** | | |
| Agriculture & Unskilled manual | Reference | |
| Skilled manual & Sales services | 30.80 (6.95, 136.57) | <0.001 |
| Clerical & Professional/ technical/ managerial | $7 \times 10^9$ (0.000) | 0.997 |
| **Income (in Myanmar Kyats)** | | |
| < 108,000 (60 USD) per month | Reference | |
| 108,000–200,000 (60–111 USD) per month | 1.77 (0.65, 4.77) | 0.262 |
| > 200,000 (> 111 USD) per month | 11.55 (2.28, 58.58) | 0.003 |
| **Religion** | | |
| Buddhism | Reference | |
| Islam | 0.29 (0.03, 2.93) | 0.295 |
| Christian | 2.24 (0.29, 17.38) | 0.441 |
| **Marital Status** | | |
| Single | Reference | |
| Married | 0.90 (0.20, 4.11) | 0.896 |
| Divorced/ Widow | 0.46 (0.06, 3.69) | 0.464 |
| **Health Related Sector** | | |
| No | Reference | |
| Yes | 0.70 (0.15, 3.24) | 0.645 |
| **COVID-19 and Vaccine-Related Experience** | | |
| **History of COVID-19 infection in respondent or within the family** | | |
| No | Reference | |
| Yes | 1.46 (0.18, 11.53) | 0.722 |
| **Hospitalization due to COVID-19 in respondents within the family** | | |
| No | Reference | |
| Yes | 0.00 (0.00) | 0.999 |
| **Severe illness or death within the family** | | |
| No | Reference | |
| Yes | $1 \times 10^9$ (0.00) | 0.999 |
| **History of COVID-19 vaccination in respondent or within the family** | | |

*(Continued)*

**Table 8.** (Continued)

| Influencing factors | COR (95% CI) | p-value |
|---|---|---|
| No | Reference | |
| Yes | 9.88 (2.46, 39.73) | 0.001 |
| **History of appearing side effects of COVID-19 vaccine in respondents or within the family** | | |
| Yes | Reference | |
| No | 2.79 (1.05, 7.42) | 0.040 |

COR, crude odds ratio; CI, confidence interval; the significance level was set at p-value <0.05.

As per the primary objective of the study, an overall acceptance level of 91.3% was observed. It was consistent with the projection of Institute for Health Metrics and Evaluation (IHME) as it estimated that less than 10% of vaccine hesitancy was existing in the Myanmar adult population, and vice versa, more than 90% of vaccine acceptance rate [35]. The vaccine acceptance rate in Myanmar was higher than in its neighbouring countries such as Bangladesh, India, and China with an overall acceptance rate of 61%, 79% and 82% respectively [9]. Among the South-East Asia (SEA) region, Myanmar had the second-highest acceptance rate of COVID-19 vaccines next to Vietnam with a 97% acceptance rate, however, currently, no studies are available for Laos, Thailand and Singapore [9]. The higher acceptance level in Myanmar compared to other countries may be due to the difference in the study population and time of the study contrasting with the time of rolling out of national vaccination programs. The current study was conducted more than one year after the vaccination program was initiated in Myanmar.

Among the total (n = 276) participants included in this study, 96.7% (267) were previously immunized with COVID-19 vaccines, but only 17.6% (47) suffered minor side effects from vaccines. Since the national COVID-19 vaccination program in Myanmar was started in January 2021, there was no announcement from MOH Myanmar on the appearance of major adverse events (MAE). Therefore, the current progress of vaccination coverage and no occurrence of MAE of vaccines comprehends the high vaccine acceptance rate of 91.3% in this study.

The age was not statistically associated with vaccine acceptance in this study as per the result of the Chi-square test (p-value = 0.103). The finding was consistent with similar studies in Indonesia and Malaysia [21, 36]. However, the study in Hong Kong showed a significant association between age and vaccine acceptance of participants [29]. This may be due to different age group classifications used in their study for their analysis (18–44, 45–64, 65 and above). Nevertheless, by observing the absolute number and percentage of acceptance rate within each group, the vaccine acceptance rate was generally increased with an increase in the age of the population consistent with similar studies [21, 29, 36]. Sex was not associated with vaccine acceptance in this study. Similarly, other studies also reported that sex was not statistically associated with willingness to receive a COVID-19 vaccine [21, 29, 36]. Conversely, significant associations were identified in some studies resulting in females being unlikely to accept vaccines than males in the US and vaccine hesitancy was higher among the female population in France [31, 37].

Although the urban population had 2.3 times higher vaccine acceptance than their rural counterparts, the effect of urbanicity had no statistically significant association in this study (p-value = 0.059). Reiter, Pennell and Katz revealed urban adults in the US had higher vaccine acceptance than rural adults with a significant association [31]. However, the study in Bangladesh reported that the rural population was significantly more likely to accept COVID-19

vaccines than urban people [38]. Nevertheless, a study amongst LMICs including Brazil, Malaysia, Thailand, Bangladesh and African nations informed no statistically significant association between residential settings (rural, suburban and urban) and the willingness to accept vaccines [39].

The education level of participants had a statistically strongly significant association with vaccine acceptance in this study. The participants with education levels of middle school and higher education were 82 times and 145 times more likely to accept vaccines than those in primary school or below (p-value < 0.001). Several studies also reported that higher education level was also associated with higher vaccine acceptance compared to the lowest education category [31, 33, 36, 38, 39]. Conversely, the study among the Indonesian population revealed that a lower acceptance rate was observed in higher education groups than in the lowest education level of junior school although the effect was not significant in their study [21].

The type of occupation of the participants was one of the independent factors associated with vaccine acceptance in this study. The participants working in skilled manual & sales services were 30 times higher vaccine acceptance than the reference group of people in agriculture & unskilled manual occupation (p-value < 0.001), however, no significant association was observed in the highest group of clerical, professional/ technical/ managerial jobs (p-value = 0.997). Wong *et al.* also mentioned a significant effect of retired people and housewives had higher vaccine acceptance than currently employed persons [29]. In addition, Harapan *et al.* informed no significant association between type of occupation and vaccine acceptance concerning the reference groups of civil servants to the other groups of private sector employees, entrepreneurs, students and retired [21]. The variation in results of association may be due to the different types of occupation used in each study while the current study applied the classification of occupation used in DHS Myanmar [40].

The monthly income was significantly associated with vaccine acceptance in this study. Compared to the reference group of participants receiving less than 108,000 MMK (basic minimum income), people receiving more than 200,000 MMK (111 USD) had 11 times more likely to receive vaccines (p-value = 0.003) while no association was seen among the middle group of people earning the salary of between 108,000 and 200,000 MMK (60–111 USD). Similarly, other studies reported a significant effect of higher income associated with higher vaccine acceptance [31, 39]. Nevertheless, no significant association between income and vaccine acceptance was reported in other studies [21, 36, 38]. The marital status of participants was not associated with vaccine acceptance in this study. A similar finding was also observed in other studies [21, 29].

While Buddhism was the major religion of most of the participants, the other religion of Islam and Christianity were also identified in this study. According to the results of both, the Chi-square test and bivariate logistics regression, religion was not significantly associated with people's acceptance of vaccines in this study. The study in Indonesia similarly reported no significant association between religion and vaccine acceptance while Islam was the major religion in their study [21].

The current study identified that the status of studying or working in the health-related sector was not associated with vaccine acceptance. Harapan *et al.* and Bono *et al.* also reported similar results of no association in their studies [21, 39]. Nevertheless, Al-Metwali *et al.* informed that healthcare workers were significantly more willing to accept vaccines than the general populace relating to their higher perceived risk of infection [22]. Another study in Hong Kong reported that suboptimal vaccination intake among general nurses while higher vaccination intention was observed in nurses who were assigned to COVID-19-related demands [41]. In addition, as the current study was a community-based study, the health

workers were not purposively selected to enter the study, and only 6.2% of participants from the health-related sector were included in this study.

In this section, it was found that all participants heard about COVID-19 and vaccines related to the current global pandemic situation of the disease. Although the participants experiencing COVID-19 infection by themselves or within household members were nearly 1.5 times more likely to accept vaccines than those without the experience, the effect of association was not statistically significant in this study (p-value = 0.722). Lazarus *et al.* and Mohamed *et al.* reported that the history of COVID-19 infection by participants or family members had no significant association with vaccine acceptance [7, 36]. On the other hand, Bono *et al.* informed that COVID-19 infection status had a significant association with willingness to receive the vaccine [39]. Reiter, Pennell and Katz stated that participants with a personal history of COVID-19 diagnosis and family member/ friend diagnosed with the infection had significantly higher vaccine acceptance than those without the history [31]. The experience of hospitalization and severe illness or death within the family had an association with the willingness to accept vaccines. Participants with previously received COVID-19 vaccination by themselves or within the family had approximately 10 times more likely to accept vaccines than those who did not previously receive it. However, respondents who appeared with the side effects of COVID-19 vaccines by themselves or family members in their previous vaccination were nearly 3 times less likely to accept future vaccination.

Among health belief model domains, regarding the perceived risk of infection or perceived susceptibility, most of the participants in this study recognized that they had a higher risk of being infected with COVID-19 (41–60% risk of infection and higher) while only 8.5% believed very low and no infection risk. The perceived risk of infection was associated with willingness to vaccination in this study (p-value <0.001). According to Al-Metwali *et al.*, more than 70% of study participants in Iraq agreed to a higher risk of COVID-19 infection [22]. Moreover, Mahmud *et al.* reported that the respondents who agreed with higher chance of being infected with COVID-19 were approximately 2 times more likely to accept vaccines than those who disagree with it [42].

When the participants were asked on whether they ever heard the COVID-19-related information from media (newspapers, pamphlets, public speakers, billboards and social media such as Facebook), there was still (0.7%) of them, especially those from the rural area did not hear about the information. Although MOH released the COVID-19-related health education and updated information on the official website (www.mohs.gov.mm), the social media page (www.facebook.com/MinistryOfHealthMyanmar), regularly sends messages to citizens through their mobile phones, and the implementation of COVID-19 awareness mask and face shield campaigns in the different regions of the country, a minor proportion of rural residents still did not reach by these activities. Other innovative approaches were mandatory to reach the COVID-19 and vaccine-related information to the whole population from the urban and rural areas of the country.

This study also revealed that other elements of HBM such as perceived benefits, subjective norms (psychological effects) and cues to action were strongly associated with vaccine acceptance of participants (Chi-square p-value <0.001). Regarding perceived benefits, most respondents agree with the statement of feeling safe and protected after the COVID-19 vaccination (84%), vaccines are effective in reducing the COVID-19 infection (80.4%) and vaccines are effective in reducing disease severity and mortality (92.4%). In subjective norm, most participants were more likely to accept after vaccination of friends. Concerning cues to action, most of the participants agree to accept vaccines upon media recommendation and endorsement of healthcare workers. Surprisingly, although there was a minor proportion of disagreed responses in other statements, there was no participant who disagree with the statement, "you

will be more likely to accept vaccine when health workers/ MOH/ WHO recommend". Al-Metwali *et al.* reported the perceived benefits, subjective norms and cues to action were influencing the willingness to take vaccines [22]. Furthermore, Mahmud *et al.* and Wong *et al.* deliberated that participants who agree with the statements of perceived benefits, subjective norms and cues to action were more likely to accept vaccines than disagree groups [29, 42]. Lastly, although different classifications of subgroups with variation in the analysis were performed in different studies such as a 5-point Likert scale or response scale of 1–6 with calculating the mean, and/ or logistics regression of agree and disagree subgroups [22, 29, 42, 43], the studies informed the significant effect of domains of HBM on the acceptance of vaccines while only Wong *et al.* reported no significant association of perceived susceptibility in acceptance of vaccines [29].

The two barriers to acceptance of COVID-19 vaccination were identified in this study. The barriers were mistrust of the efficacy of vaccines (75%) and potential major adverse events of COVID-19 vaccines (25%). MacDonald *et al.* mentioned that delay or refusal of vaccination was influenced by multiple contextual factors, personal perception of the vaccines and particular concerns associated with vaccine or vaccination. The cultural, social, emotional, and political factors and context of vaccines were determinants of vaccine hesitancy [44].

The quality control including manufacturing and cold-chain storage, side effects of vaccines, perceived less severity of COVID-19 infection, postpone vaccination for another year until tested by others, doubt on vaccine effectiveness and preference for natural immunity were concerns related to COVID-19 vaccine hesitancy [8]. Paul, Steptoe and Fancourt mentioned that doubt on vaccine benefits and distress on unforeseen long-term side effects were the most important determinants related to uncertainty and reluctance to vaccinate against COVID-19 among the UK adult population [45]. Al-Metwali *et al.* highlighted the three main barriers to vaccine acceptance by the general population and healthcare workers in their study and reported distress in storage conditions (84.7%), and adverse events (62.6%) and efficacy of vaccines (44.5%) were major barriers [22].

As per results of the study, the community acceptance of COVID-19 vaccines was 91.3% which built the confidence for public health officials to reach the high immunization coverage within the community in line with global COVID-19 vaccination strategy [46]. Nevertheless, as per WHO quote of "no one is safe from COVID-19 until everyone is safe", the MOH required to provide complete vaccination services to everyone within the community [47]. In order to increase COVID-19 vaccine acceptance within the community in Myanmar, a multifaceted approach is required. Firstly, it is essential to upsurge public education by providing precise and accessible information about the vaccine. Public health officials could use various media platforms to communicate clear, consistent messages about the vaccine's safety and efficacy, tailored to address specific concerns or myths that people may have.

While the study found that people who are uncertain of the effectiveness of vaccines and concern about major adverse effects are more likely to refuse vaccinations, the issue of vaccine hesitation has to be tackled head-on by comprehending the underlying causes. In addition to listening to people's concerns, healthcare professionals and community leaders may give accurate information and their own experiences with the vaccination. To guarantee that everyone who wants the vaccination may receive it, access to it is essential. This can include establishing immunization clinics in community centres, churches, and educational institutions as well as providing transportation for individuals who require it to vaccination sites.

Influential community members such as community leaders, religious figures, and healthcare professionals can promote vaccination and share their personal experiences with the vaccine because the public is more likely to trust the opinion of someone they know and respect. It is critical to address the challenges marginalized communities encountered in obtaining

vaccines, such as transportation issues, language problems, and mistrust of the healthcare system. To guarantee equal vaccination delivery, efforts should be done.

Last but not least, continuing assistance is required to address concerns and encourage continued vaccination uptake. After receiving the vaccination, some people may have adverse effects or anxiety about it; in these cases, continued support from medical professionals or community members can help relieve these worries. By these means, the MOH can boost community acceptability of the COVID-19 vaccination by putting these steps into practice.

## Limitations of study

Our study has some limitation. Firstly, it was not possible to conduct a mixed method design in this study due to limited resources and time frame. This quantitative design with questionnaires limited the freedom of participants to explore further influencing factors of the acceptance of vaccines such as the origin of vaccines, trust in healthcare workers, political issues, personal perception and motivation for vaccination, social procedures and norms, and practical factors to fully understand the behavioural and social drivers of vaccination. The study area included the townships within Rakhine state (out of 14 states/ regions and one territory) of Myanmar. Even using a proper sampling procedure with adequate sample size, the results of this research would reflect the community in Rakhine state and could not generalize to the other population from different regions of the country.

## Conclusion

Since the current study was the primary and original research in Myanmar which identified the vaccine acceptance rate (91.3%) of the population, and further explored the associated factors for vaccine acceptance and barriers to vaccination, the findings of the study provided valuable evidence for public health authorities to estimate the possibility and success of current COVID-19 vaccination program as well as to implement the strategic interventions upon the specific group of population to increase the COVID-19 vaccine uptake. Furthermore, health authorities could use the results of the study as a baseline, and then assess the acceptance level after implementing vaccine promotion campaigns to monitor the changes in vaccine acceptance of the population over time and evaluate the effectiveness of specific vaccine promotion activities. The study implicated the crucial need for education campaigns to the specific group population to upsurge vaccine acceptance. According to the results of the study, the healthcare workers of MOH should frequently communicate with the community to tackle the misinformation on COVID-19 vaccines and to understand the importance of vaccination in the control of the pandemic. During the process, community engagement is fundamental while interacting with local leaders and community-based organizations would fully understand the perceptions of the community and develop an effective vaccine promotion activity to achieve the highest achievable level of vaccination coverage within the community.

## Supporting information

**S1 Fig. Health belief model elements and variables which have the potential to influence people's acceptance of receiving a COVID-19 vaccine (Source: Al-Metwali et al., 2021).** (TIF)

## Acknowledgments

I would like to express sincere and heartfelt thanks to professors and research supervisors from both the University of Bedfordshire, UK and STIMU, Myanmar, the State Health

Director and Deputy State Health Director from the State Health Department of Rakhine State, and the Field Coordinators of COVID-19 Response Activity from Myanmar Health Assistant Association (MHAA) and Myanmar Nurse and Midwife Association (MNMA) for their academic, technical, psycho-social support and other necessary support throughout the entire research study.

## Author Contributions

**Conceptualization:** Saw Simon, Kyaw Myo Tun.

**Data curation:** Saw Simon, Kaung Myat Min, Tun Zaw Latt, Pa Pa Moe, Kyaw Myo Tun.

**Formal analysis:** Saw Simon, Kyaw Myo Tun.

**Methodology:** Saw Simon.

**Project administration:** Saw Simon, Kyaw Myo Tun.

**Software:** Saw Simon, Kyaw Myo Tun.

**Supervision:** Kaung Myat Min, Tun Zaw Latt, Pa Pa Moe, Kyaw Myo Tun.

**Validation:** Saw Simon, Tun Zaw Latt, Pa Pa Moe.

**Visualization:** Saw Simon.

**Writing – original draft:** Saw Simon.

**Writing – review & editing:** Saw Simon, Kyaw Myo Tun.

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
