## [Decision Letter · Decision Letter 0]

6 Mar 2023

PGPH-D-22-01528

The Community Acceptance of COVID-19 Vaccines in Rakhine State: A Cross-sectional Study in Myanmar

Dear Dr. Simon,

Thank you for submitting your manuscript to PLOS Global Public Health. After careful consideration, we feel that it has merit but does not fully meet PLOS Global Public Health’s publication criteria as it currently stands. Therefore, we invite you to submit a revised version of the manuscript that addresses the points raised during the review process.

We look forward to receiving your revised manuscript.

Kind regards,

Jianhong Zhou

Staff Editor

Journal Requirements:

Additional Editor Comments (if provided):

Reviewers' comments:

Reviewer's Responses to Questions

**Comments to the Author**

1. Does this manuscript meet PLOS Global Public Health’s publication criteria? Is the manuscript technically sound, and do the data support the conclusions? The manuscript must describe methodologically and ethically rigorous research with conclusions that are appropriately drawn based on the data presented.

Reviewer #1: Yes

Reviewer #2: Yes

2. Has the statistical analysis been performed appropriately and rigorously?

Reviewer #1: Yes

Reviewer #2: Yes

3. Have the authors made all data underlying the findings in their manuscript fully available (please refer to the Data Availability Statement at the start of the manuscript PDF file)?

Reviewer #1: Yes

Reviewer #2: Yes

4. Is the manuscript presented in an intelligible fashion and written in standard English?

Reviewer #1: Yes

Reviewer #2: Yes

5. Review Comments to the Author

Reviewer #1: Thanks for the invitation to review this manuscript.

In the current study, the authors investigated the issue of COVID-19 vaccine acceptance and its associated determinants in Rakhine State, Myanmar.

The importance of this study can be related to the previous scarcity of such reports from Myanmar as highlighted previously in the recent review: https://doi.org/10.2147/JMDH.S347669, cited by the authors.

Therefore, the current study present original data on a timely and important public health issue despite the decline in interest in the research involving COVID-19 vaccine hesitancy.

The acceptance rate of COVID-19 vaccination in the study was high, which was in line of the results from countries in South East Asia.

Overall, the manuscript is well written with comprehensive literature review. The methodology is valid and the results were presented clearly and supported the conclusions.

Importantly, the authors elaborated well on the potential limitations of the study.

Therefore, I think that the manuscript can be considered for publication.

Congratulations!

Reviewer #2: Introduction/Background: Consider deleting or editing down lines 73-98 and 108-179. While these sections are well-written and informative, the paper would benefit from a tighter focus on Myanmar itself. Consider replacing this language with additional details on Myanmar's public health infrastructure, vaccination challenges, etc.

Line 190: Figure 1 appears to be missing from the manuscript.

Re: additional figures: Consider including a map of Myanmar that highlights where the study was conducted.

In the methods section, please explain what the health belief model is, and why it was chosen to analyze the findings from this study.

Participants (Line 284): Please define "household leader" - does this refer to a male head of household?

Discussion: Please elaborate on how the findings of this study could inform vaccine delivery, risk communication, and/or community engagement around vaccination in Myanmar. In other words, what implications does this study have for public health policymaking and practice?

I would recommend moving the limitations to the discussion section instead of listing them after the conclusion. Additionally, consider commenting on the health belief model itself -- do you believe it is a sufficient tool for studying vaccine acceptance?

6. PLOS authors have the option to publish the peer review history of their article (what does this mean?). If published, this will include your full peer review and any attached files.

**Do you want your identity to be public for this peer review?** For information about this choice, including consent withdrawal, please see our Privacy Policy.

Reviewer #1: No

Reviewer #2: No

---

## [Decision Letter · Decision Letter 1]

17 Apr 2023

PGPH-D-22-01528R1

The Community Acceptance of COVID-19 Vaccines in Rakhine State: A Cross-sectional Study in Myanmar

Dear Dr. Simon,

Thank you for submitting your manuscript to PLOS Global Public Health. After careful consideration, we feel that it has merit but does not fully meet PLOS Global Public Health’s publication criteria as it currently stands. Therefore, we invite you to submit a revised version of the manuscript that addresses the points raised during the review process.

We look forward to receiving your revised manuscript.

Kind regards,

Julio Croda, Ph.D, M.D.

Academic Editor

Journal Requirements:

Additional Editor Comments (if provided):

Reviewers' comments:

Reviewer's Responses to Questions

**Comments to the Author**

1. If the authors have adequately addressed your comments raised in a previous round of review and you feel that this manuscript is now acceptable for publication, you may indicate that here to bypass the “Comments to the Author” section, enter your conflict of interest statement in the “Confidential to Editor” section, and submit your "Accept" recommendation.

Reviewer #1: All comments have been addressed

Reviewer #3: All comments have been addressed

2. Does this manuscript meet PLOS Global Public Health’s publication criteria? Is the manuscript technically sound, and do the data support the conclusions? The manuscript must describe methodologically and ethically rigorous research with conclusions that are appropriately drawn based on the data presented.

Reviewer #1: Yes

Reviewer #3: Partly

3. Has the statistical analysis been performed appropriately and rigorously?

Reviewer #1: Yes

Reviewer #3: Yes

4. Have the authors made all data underlying the findings in their manuscript fully available (please refer to the Data Availability Statement at the start of the manuscript PDF file)?

Reviewer #1: Yes

Reviewer #3: Yes

5. Is the manuscript presented in an intelligible fashion and written in standard English?

Reviewer #1: Yes

Reviewer #3: Yes

6. Review Comments to the Author

Reviewer #1: The manuscript provides an accurate and detailed analysis of the research conducted. Therefore, I endorse the manuscript for publication. Best wishes!

Reviewer #3: The study presents primary and original data from the region, and the methodology, despite having some limitations, allows the authors to achieve their proposed objectives. However, the text contains many words, tables, and figures, making it time-consuming to read. Therefore, the authors should consider removing some of the unnecessary explanations from the introduction and methodology sections, which would help the reader focus on the main content and make the reading more fluid. Additionally, the limitations should be restated since they have an impact on the results.

Below are some suggestions and questions for the authors:

Line 41 - It would be helpful to convert the monthly income to US dollars to enable readers from other parts of the world to understand.

Lines 90 to 159 - The information presented in this section doesn't add solidity to the manuscript. If the authors consider it important, they could present it as supplementary material. Alternatively, some of the information presented could be used in the introduction to make the reading more fluid and focused on the objective of the work.

Research Design - The explanation of the Health Belief Model (HBM) seems unnecessary. The authors can cite references on the subject or consider it as supplementary material. This section should be objective, with the authors only stating what type of design was used in the study. The text from line 185 must be included in the results section.

Line 219 - It's unclear what "N4Studies" means.

Sampling Procedure - The text contains many justifications and explanations. The authors should only say how the sampling process was carried out, as between lines 237-240. This is one of the main limiting aspects of the study, which should be highlighted in the section "Limitations of Study" (line 738), which has been deleted.

Participants - The authors should rename this section to "Eligibility Criteria" and clearly state who was included in the study (and the criteria followed) and who was excluded (and the criteria followed).

Data Collection Tool - Were the questionnaires printed or completed on a mobile device? If printed, how were the data digitized after answering the questions? Was any quality control procedure adopted for this typing? Where were the completed questionnaires stored?

Ethical Considerations - The authors should limit themselves to the ethical approvals obtained. The text between lines 273-278 should be deleted. If essential for methodological understanding, explanations of how the interviews were carried out (lines 284-289) should not appear in this section. If ethical approval opinions have any identifiers such as a numeric or alphanumeric code, they should be presented.

Data Entry and Analysis - Were the questionnaires printed or completed on a mobile device? If printed, how were the data digitized after answering the questions? Was any quality control procedure adopted for this typing? Where were the completed questionnaires stored?

Line 305 - It's unclear why 276 participants were included when the sample calculation (line 221) predicted 288.

Line 349 - The authors need to add the unit of measurement ("108,000 and 200,000 per month").

Table 1 - The "Income" unit of measurement could be in US dollars to enable readers from other regions of the world to understand better. In the "Health Related Sector" variable, only the "No" answer should be presented.

Lines 329/330 - The text is repetitive and has already been mentioned in the "Data Entry and Analysis" section. Therefore, it should be removed.

Influencing Factors for the Acceptance of COVID-19 Vaccines - The authors should incorporate these data into the section "Factors Associated with Acceptance of COVID-19 Vaccines."

In the original version the authors present a limitations section (lines 738-747), which does not appear in the current version. The study presents original and relevant data for the region, but it has limitations that should be highlighted, so that the reader can contextualize the findings.

7. PLOS authors have the option to publish the peer review history of their article (what does this mean?). If published, this will include your full peer review and any attached files.

**Do you want your identity to be public for this peer review?** For information about this choice, including consent withdrawal, please see our Privacy Policy.

Reviewer #1: **Yes: **Malik Sallam

Reviewer #3: **Yes: **Roberto D Oliveira

<quillbot-extension-portal></quillbot-extension-portal>

---

## [Decision Letter · Decision Letter 2]

20 Jun 2023

The Community Acceptance of COVID-19 Vaccines in Rakhine State: A Cross-sectional Study in Myanmar

PGPH-D-22-01528R2

Dear Mr Simon,

We are pleased to inform you that your manuscript 'The Community Acceptance of COVID-19 Vaccines in Rakhine State: A Cross-sectional Study in Myanmar' has been provisionally accepted for publication in PLOS Global Public Health.

Best regards,

Julio Croda, Ph.D, M.D.

Academic Editor

Reviewer Comments (if any, and for reference):

Reviewer's Responses to Questions

**Comments to the Author**

1. If the authors have adequately addressed your comments raised in a previous round of review and you feel that this manuscript is now acceptable for publication, you may indicate that here to bypass the “Comments to the Author” section, enter your conflict of interest statement in the “Confidential to Editor” section, and submit your "Accept" recommendation.

Reviewer #3: All comments have been addressed

2. Does this manuscript meet PLOS Global Public Health’s publication criteria? Is the manuscript technically sound, and do the data support the conclusions? The manuscript must describe methodologically and ethically rigorous research with conclusions that are appropriately drawn based on the data presented.

Reviewer #3: Yes

3. Has the statistical analysis been performed appropriately and rigorously?

Reviewer #3: Yes

4. Have the authors made all data underlying the findings in their manuscript fully available (please refer to the Data Availability Statement at the start of the manuscript PDF file)?

Reviewer #3: Yes

5. Is the manuscript presented in an intelligible fashion and written in standard English?

Reviewer #3: Yes

6. Review Comments to the Author

Reviewer #3: All suggestions were accepted by the authors, as well as all questions were answered.

The main text incorporates a number of changes.

7. PLOS authors have the option to publish the peer review history of their article (what does this mean?). If published, this will include your full peer review and any attached files.

**Do you want your identity to be public for this peer review?** For information about this choice, including consent withdrawal, please see our Privacy Policy.

Reviewer #3: No

<quillbot-extension-portal></quillbot-extension-portal>